# Modeling User Reputation in Online Social Networks: The Role of Costs, Benefits, and Reciprocity

**DOI:** 10.3390/e22101073

**Published:** 2020-09-24

**Authors:** Frank Schweitzer, Pavlin Mavrodiev, Adrian M. Seufert, David Garcia

**Affiliations:** 1Chair of Systems Design, ETH Zurich, Weinbergstrasse 58, 8092 Zurich, Switzerland; pmavrodiev@ethz.ch (P.M.); aseufert@ethz.ch (A.M.S.); 2Complexity Science Hub Vienna, Josefstädter Strasse 39, 1080 Vienna, Austria; garcia@csh.ac.at; 3Section for Science of Complex Systems, CeMSIIS, Medical University of Vienna, Spitalgasse 23, 1090 Vienna, Austria

**Keywords:** reciprocity, core-periphery network, cost-benefit relation, robustness, reputation

## Abstract

We analyze an agent-based model to estimate how the costs and benefits of users in an online social network (OSN) impact the robustness of the OSN. Benefits are measured in terms of relative reputation that users receive from their followers. They can be increased by direct and indirect reciprocity in following each other, which leads to a core-periphery structure of the OSN. Costs relate to the effort to login, to maintain the profile, etc. and are assumed as constant for all users. The robustness of the OSN depends on the entry and exit of users over time. Intuitively, one would expect that higher costs lead to more users leaving and hence to a less robust OSN. We demonstrate that an optimal cost level exists, which maximizes both the performance of the OSN, measured by means of the long-term average benefit of its users, and the robustness of the OSN, measured by means of the lifetime of the core of the OSN. Our mathematical and computational analyses unfold how changes in the cost level impact reciprocity and subsequently the core-periphery structure of the OSN, to explain the optimal cost level.

## 1. Introduction

Online social networks (OSN), like other types of social organizations, undergo a steady evolution. This is not only caused by the behavior of users while using the network [1], but even more if new users enter the network, while other users may decide to leave. New links between users are formed online, based on shared information, friendship, common interests, etc., and existing links may be deleted if users leave, or commonalities have changed. Under normal circumstances, such events may not jeopardize the existence of the OSN, in particular if the OSN is still popular and growing. However, losing users and links can pose a serious risk to OSN, even for large and successful ones on the scale of Twitter and Facebook. As a recent empirical study [2] has shown, the dropout of some users can trigger cascades of other users leaving, which quickly accumulates to a level that threatens the existence of the OSN.

To investigate the emergence and the impact of such cascades analytically and numerically, we propose a model that explicitly considers the costs and benefits of users of an OSN. *Benefits* are measured in terms of relative *reputation* (see Section 2.1), that is we take into account (i) the *number* and the *direction* of links between users, and (ii) the impact of the reputation of the counterparts on the reputation of a given user. A gain of reputation can be seen as a benefit of being part of a social network. However, there are also *costs* involved in being a member of an OSN, even if they are implicit. These include, among others, the efforts to login, to maintain the individual profile, to learn how to efficiently use the GUI of the provider, or the effort to adapt to changes in the GUI.

Our model extends previous notions of reputation in social networks which are based on the degree or the centrality of a user, measured by traditional centrality metrics [3]. In contrast to such topological metrics, we define the reputation of a user based on the reputation of those users that are linked to it. That means there is a *value* assigned to incoming and outgoing links, as they increase either the own reputation or that of others. This motivates us to consider (i) a *directed network* and (ii) a self-consistent *dynamics* of how reputation changes with the reputation of others.

The assumption that individual reputation increases with the reputation of connected users is quite common in different ranking schemes. For example, the earliest version of Google’s
PageRank algorithm calculates the rank of a website as a function of the ranks of the sites linking to it. Similarly, the vulnerability of financial institutions can be calculated dependent on the vulnerability of the counterparties connected to it [4]. In more general terms, such an assumption follows (hetero)catalytic models of prebiotic evolution or foodweb interactions, e.g., from chemistry and biology, where the concentration of a particular (chemical) species depends on the concentration of those species that produce, or feed, it [5].

Jain and Krishna [6] have combined these dynamics with a network dynamic that runs on a different time scale. At each time step (measured in network time), the system was perturbed by an extremal dynamic where the least performing node in the network was replaced by a new node that randomly rewires itself back to the system. This model was already analyzed in detail [7,8] and extended to cover other phenomena, such as strategic link formation between economic agents [9].

In this paper, we build on the existing model class, but extend it in several important points: (i) In contrast to the simple extremal dynamics (where only one user is removed), we introduce a condition for users to decide to leave, namely we allow *many* more users to leave dependent on their *personal* cost-benefit ratio. (ii) We focus on the effects that cascades of users leaving have on the robustness and the performance of the OSN. *Performance* is quantified using the long-term average reputation of users (see also Section 3.1) and is taken as a systemic measure for the OSN rather than as an individual one. *Robustness* implies that the decision of users to leave the OSN does not trigger large cascades of other users leaving. We will proxy this by means of the lifetime of the core of the OSN (see also Section 2.3).

Our aim is not to focus on the *size* of cascades, which was done in different network approaches to *systemic risk* [10,11], but rather to understand the impact that leaving users have on the OSN. As mentioned, users will leave if their costs exceed their benefits. Different from previous publications we *vary* this cost, which is the crucial parameter in our model. Interestingly, we find that both the robustness and the performance of the OSN are maximized for a *non-zero cost* of usage.

To explain this, we must investigate how directed links impact the reputation across the OSN. This leads us to the problem of *reciprocity*: If a user contributes to the reputation of *others*, she may expect that these users in return also increase her own reputation. Mutual directed links between two users, e.g., 1→2→1, would indicate *direct reciprocity*. However, users can also indirectly increase their reputation if they are part of a *cycle*, e.g., 1→2→3→1, which indicates *indirect reciprocity*.

While direct and indirect reciprocity is advantageous for the OSN from a user perspective, it is not well understood how reciprocity adds to systemic properties such as the robustness of the OSN. We will address this question by a mathematical analysis that explains the impact such cycles on the core-periphery structure of the OSN. Users that are part of a cycle are well integrated in the OSN, maintain a high reputation and have little incentives to leave. However, if users leave because of a negative cost-benefit ratio, this will impact the number and sizes of such cycles, which we need to understand by means of an analytic approach that is complemented by computer simulations.

Our paper is organized as follows: Section 2 introduces the model and the analytical framework to analyze the structure of an OSN. Different Appendices allow the study of this framework by following concrete step-by-step examples. In Section 3 we present the results of computer simulations, to demonstrate the emergence of an optimal cost value, and we discuss the impact of different model parameters on the structure and the performance of the OSN. A discussion of the general insights and concluding remarks are provided in Section 4.

## 2. A Reputation Model

### 2.1. Costs and Benefits

Why do users leave an online social network (OSN)? A rational answer should be, they leave because their costs of staying in the network exceed, at a given point in time, their benefits of being members. This can be expressed by the dynamics:(1)$$s_{i}\left(t\right)=\Theta \left[b_{i}\left(t\right)-c_{i}\left(t\right)\right]$$
Here, $s_{i}\left(t\right)$ characterizes the current state of user *i* at time *t* as a binary variable: $s_{i}=1$ means that the user is part of the network and $s_{i}=0$ means that at time *t* the user leaves. Θ[z] is the Heavyside function which returns 1 if z≥0 and 0 if z<0. Thus, the current state of user *i* depends on the difference between its benefits $b_{i}\left(t\right)$ and costs $c_{i}\left(t\right)$.

In the following, we assume that the benefits of a user to join, and to stay, in the OSN result from the *reputation*
$X_{i}\left(t\right)$ that the user receives from being connected to other users in the OSN. There are many social networks that operate this way: in Twitter users get a reputation from the number of their followers, in product review communities like Amazon or YouTube users earn their reputation from the votes of other users. In our model, we particularly assume that the reputation of a user, *i*, does not just depend on the sheer number of other users that follow *i*, but also on their reputation. In other words, if a user *j* with a high reputation $x_{j}\left(t\right)$ connects to user *i*, the latter receives more reputation than from a follower with low reputation.

Such a reputation measure can be explicitly displayed on the site, like the Reddit karma or the *RG score* of Researchgate, which increases with the reputation of followers and the feedback of the community. On the other hand, user reputation can be implicit and not part of a user profile, but can still be perceived through the activity of other users. Examples of this implicit reputation are *retweets* in Twitter and *likes* in Facebook.

We can express the impact of the followers on the reputation of a user by the following dynamics:(2)$$\frac{dX_{i}}{dt}=\sum_{j=1}^{N}a_{ij}\left(t\right)X_{j}\left(t\right)-\phi X_{i}\left(t\right)$$
The coefficients $a_{ij}$’s are elements of the adjacency matrix of the OSN, **A**. They represent the link between users *j* and *i* in the OSN at time *t*. These are unweighted, but directed links, because it makes a difference whether user *j* follows user *i*, or the other way round. If $a_{ij}\left(t\right)=1$ then there is a link from *j* to *i*, in which case *j* is called a *follower* of *i*. $a_{ij}\left(t\right)=0$ indicates the absence of directed connection from *j* to *i*. Since a user cannot follow herself, we set $a_{ii}\left(t\right)=0$ for all *t*.

The sum in Equation (Equation 2) is over all users that can potentially link to *i*. That means if several users leave the OSN, they are replaced by the same number of new users joining the OSN. This way the total number of users, *N*, is kept constant. This is a first, and not necessarily the most realistic, approximation to consider an *entry and exit* dynamics, which can be refined in subsequent investigations. The second term on the r.h.s. of Equation (Equation 2) takes into account the effort to keep a certain level of reputation. Without several followers, the reputation of user *i* cannot be maintained and thus decays exponentially over time with a rate ϕ. The larger ϕ, the higher the effort to maintain the reputation.

To relate reputation to the benefits of users in the OSN, we could simply assume that benefits grow in proportion to reputation. On the other hand, the costs to stay in the OSN can be assumed to be a constant τ equal for all users, which can be seen as the effort to login and stay active. Please note that there is no cost involved in maintaining links. According to Equation (Equation 2) and dependent on the specific social network, the reputation of users can grow to large numbers, while on the other hand it can become infinitely small, but never zero. Because we are more interested in the reputation of users *relative* to others, we will rescale the benefits from the reputation by the largest value $X_{\max}\left(t\right)$, which makes different networks more comparable. Hence, the benefits and costs are specified in this model as follows:(3)$$b_{i}\left(t\right)=\frac{X_{i}\left(t\right)}{X_{\max}\left(t\right)}\in \left(0,1\right)\;;\;c_{i}\left(t\right)=\tau \in \left[0,1\right)$$
Please note that in contrast to Equation (Equation 3), it is common to express the relative reputation in terms of the sum of the individual reputations, $\sum X_{i}\left(t\right)$:(4)$$x_{i}\left(t\right)=\frac{X_{i}\left(t\right)}{\sum_{j}X_{j}\left(t\right)}$$
In the context of an OSN, however, this is unrealistic, as it requires that either the total reputation or everyone’s reputation is public knowledge. For this reason, we posit that users compare their reputation to the most reputable individual, who is often visible in rankings of user reputation. We note that despite the conceptual difference, a solution to Equation (Equation 3) can be mapped directly to a corresponding solution to Equation (Equation 4), by normalizing $x_{i}$ with respect to $x_{\max}$ as follows:(5)$$\frac{x_{i}\left(t\right)}{x_{\max}\left(t\right)}=\frac{X_{i}\left(t\right)/\sum_{j}X_{j}\left(t\right)}{X_{\max}\left(t\right)/\sum_{j}X_{j}\left(t\right)}=\frac{X_{i}\left(t\right)}{X_{\max}\left(t\right)}=b_{i}\left(t\right)$$

In Appendix A, we further show that an equilibrium solution to Equation (Equation 2) is also an equilibrium for $b_{i}\left(t\right)$ and $x_{i}\left(t\right)$ up to a scaling factor. This means that the entry/exit dynamics introduced in Section 2.3 is invariant to the particular way in which users evaluate their relative reputation.

According to Equation (Equation 1), users leave the OSN at time *t* if their *relative reputation* is lower than a fixed threshold. Their links $a_{ij}\left(t\right)$ are then set to zero, which according to Equation (Equation 2) reduces the reputation of other users *j* at the next time step. This can lead to cascades of users leaving the OSN at consecutive time steps. The aim of our paper is to understand how a decrease of users’ motivation to stay because of an increase in their fixed costs τ will impact the OSN. Therefore, as a next step, in Section 2.2 we first investigate how the reputation depends on the social network, before turning to the entry/exit dynamics in Section 2.3.

### 2.2. Quasi-Stationary Equilibrium

Let us first discuss the reputation dynamics of users for a *fixed* social network. Expressing the dynamics of $b_{i}\left(t\right)$ from Equation (Equation 3) yields (see Appendix A):(6)$$\frac{db_{i}}{dt}=\sum_{j=1}^{N}a_{ij}b_{j}\left(t\right)-b_{i}\left(t\right)\sum_{j=1}^{N}a_{zj}b_{j}\left(t\right)$$
where *z* is the index of the individual with highest absolute reputation $X_{\max}$. The first term describes the reputation boost that individual *i* obtains from all her followers. The second term is a scaling factor and represents the reputation decay with strength equal to the total boost in reputation that user *z* receives.

#### 2.2.1. Eigenvalues and Eigenvectors

The set of Equations (Equation 6) forms a linear dynamical system of coupled first-order differential equations with the initial conditions described by the vector $b\left(0\right)=\{b_{1}\left(0\right),b_{2}\left(0\right),...,b_{N}\left(0\right)\}$. The (constant in time) $a_{ij}$ forms the elements of an adjacency matrix A of size N×N (see Figure 1).

At equilibrium we require $\dot{b_{i}}=0$, hence
(7)$$\sum_{j=1}^{N}a_{ij}b_{j}\left(t\right)=b_{i}\left(t\right)\sum_{j=1}^{N}a_{zj}b_{j}\left(t\right)$$
In matrix form, this equation becomes
(8)$$Ab\left(t\right)=b\left(t\right)\sum_{j=1}^{N}a_{zj}b_{j}\left(t\right)$$

If $X^{\lambda }$ is an eigenvector of A with a corresponding eigenvalue λ, then rescaling $X^{\lambda }$ will also produce an eigenvector, i.e., $b^{\lambda }=X^{\lambda }/X_{\max}^{\lambda }$ is a solution to Equation (Equation 8). In this case the scaling factor $\sum_{j}a_{zj}b_{j}\left(t\right)$ gives the zth component, ${\left(\lambda b^{\lambda }\right)}_{z}$, of the vector $\lambda b^{\lambda }$. Since ${\left(\lambda b^{\lambda }\right)}_{z}=X_{z}^{\lambda }/X_{\max}^{\lambda }=1$, it follows that $\sum_{j}a_{zj}b_{j}\left(t\right)=\lambda$.

The dynamics of Equation (Equation 8) are dominated by the largest eigenvalue of A and the corresponding eigenvector gives us the steady-state reputation values. Let us illustrate this by the didactical example of a rather small network shown in Figure 1.

The characteristic polynomial determining the eigenvalues of the corresponding adjacency matrix A given in Figure 1 is
(9)$$-\lambda^{5}+\lambda^{3}+\lambda^{2}=0$$
and the largest eigenvalue is $\lambda_{1}$=1.32. The corresponding eigenvector gives us the absolute reputation: $X^{\lambda_{1}}=\left\{2.32,1.75,1.32,1.32,1\right\}$. Rescaling this eigenvector by $1/X_{\max}^{\lambda_{1}}$ gives the relative reputation in equilibrium: $b^{\lambda_{1}}=\left\{1,0.75,0.57,0.57,0.43\right\}$.

#### 2.2.2. Core-Periphery Structure

To facilitate the interpretation of the relative reputation values in Figure 1, let us take a look at the corresponding network structure. This toy network already shows a topological feature typical for many social networks, known as a *core-periphery* structure [12,13,14,15,16,17,18]. Following Borgatti and Everett [19], Everett and Borgatti [20], the core is defined as a cohesive subgroup (e.g., a clique, *n*-clique, *n*-club or *n*-clan, *k*-plex) and the periphery is everything else.

Accordingly, and accounting for the directionality in reputation-based OSNs, we define the *core*, *Q*, as the *largest strongly connected component* (SCC) in the OSN. Each node in the SCC is reachable from all other nodes in the SCC. The *periphery* consists of all nodes that do not belong to the core. In Appendix B we characterize the structure of the code *Q* as it depends on the largest eigenvalue of the adjacency matrix of the OSN $\lambda_{1}$. In Figure 1, the core is the SCC formed by users *1*, *2*, *3*, who mutually boost each other’s reputations. The periphery consists of users *4* and *5* who only benefit from being connected to the core. The periphery usually contains simple *chains* of users (*4* → *5*) which emanate from the core, in this case from user *2*. The core itself does not contain simple chains, but *cycles*, that is *closed* directed chains that involve n≥2 users [7]. In the example, we observe two cycles, *1* →*2*→*1* and *1*→*2*→ *3* → *1*, i.e., users can be part of more than one cycle. Moreover, all users have one follower each, except for user *1* who has two followers, thus her reputation can be expected to be higher than that of the others. Please note that even though users *4* and *5* have the same number of followers, their reputation is different, as user *4* has a more reputable follower belonging to the core.

More formally, a *cycle* is defined as a subgraph in which there is a closed path from every node of the subgraph back to itself. Cycles and structures of interlocking cycles represent *irreducible subgraphs*. The core *Q* must always contain at least one cycle to qualify as a strongly connected component. In Appendix B we show the dependency between the largest eigenvalue of the adjacency matrix of the network and the length and number of the cycles in the core.

#### 2.2.3. Direct and Indirect Reciprocity

The shortest cycle, n=2, expresses *direct reciprocity*. In the example in Figure 1 users *1* and *2* mutually follow each other and boost each other’s reputations. For n≥3, we have *indirect reciprocity*, i.e., user *2* follows *3*; however, *3* does not follow *2*, but follows *1* instead, and only *1* may follow *2*, thus closing the loop.

Direct reciprocity is very common in OSN, e.g., in twitter or google+ it is seen as good practice to link back to someone that has chosen to follow you or to have you as his/her friend. Likewise, likes, +1, or shared posts often take reciprocity into account. Compared to this, indirect reciprocity is more difficult to detect. To boost interaction along a chain of followers, and to hopefully close the loop, some OSN, e.g., google+ or researchgate, indicate for each follower the number of additional users that the user and the follower both have in common. This may increase the likelihood of creating shortcuts and shorter cycles.

#### 2.2.4. Length of Simple Chains

At equilibrium, $db_{i}/dt=0$, we can insert the eigenvector $b^{\lambda_{1}}$ corresponding to $\lambda_{1}$ into Equation (Equation 6) to get
(10)$$\sum_{j}a_{ij}b_{j}^{\lambda_{1}}=\lambda_{1}b_{i}^{\lambda_{1}}\;;\;b_{i}^{\lambda_{1}}=\frac{1}{\lambda_{1}}\sum_{j}a_{ij}b_{j}^{\lambda_{1}}$$
This means that in the long run, the reputation $b_{i}$ of user *i* is equal to the sum of the reputations of all users *j* that follow *i*, *attenuated* by a factor $1/\lambda_{1}$ [6,7].

Equation (Equation 10) allows us to draw some conclusions about the maximum length of simple chains involving peripheral users. In the example discussed, we note that the reputations of users *4* and *5* are related by the attenuation factor in Equation (Equation 10) such that $b_{4}=b_{2}/\lambda_{1}$, $b_{5}=b_{2}/\lambda_{1}^{2}$, and in general $b_{n}=b_{2}/\lambda_{1}^{n-1}$ for a chain of length *n*. If we require a simple chain to be exactly of length *n*, $b_{n}>\tau$ and $b_{n+1}\leq \tau$ must hold. In other words, the nth peripheral user finds it beneficial to stay while the (n+1)th leaves. Hence, we obtain for *n*: (11)$$n\in \left[\frac{\ln(b_{2}/\tau )}{\ln\lambda_{1}},\frac{\ln(b_{2}/\tau )}{\ln\lambda_{1}}+1\right)\;;\;n=\lceil \frac{\ln(b_{2}/\tau )}{\ln\lambda_{1}}\rceil$$
since we require *n* to be an integer value. The maximum length of a simple chain in a core-periphery network thus depends on the cost level τ, the connectedness within the core *Q* expressed by the largest eigenvalue $\lambda_{1}$, and the relative reputation of the core user who connects the core to the chain (in our example user *2*).

#### 2.2.5. Unstable Cores

Imposing the condition n>0 in Equation (Equation 11) requires that $\lambda_{1}>1$ and $b_{2}>\tau$ which holds only if the OSN contains *cycles*. Without these cycles, $\lambda_{1}=0$ and the core-periphery structure breaks down. The condition, $b_{2}>\tau$, requires that the core user *2*, who connects to the simple chain, needs to obtain a net gain from participating in the OSN. Otherwise, due to the attenuation factor in Equation (Equation 10), none of the users in the chain would have an incentive to stay in the network.

The special case of $\lambda_{1}=1$ represents an important exception. In this case, and provided that $b_{2}>\tau$, the length of the simple chain is no longer bounded, since all users in the chain would have the same relative reputation. More importantly, however, network structures like these are very unstable, because core users have a reputation comparable to the periphery. In fact, peripheral users can often obtain a higher reputation from multiple connections to the core, which reduces the benefit of the core users and increases their likelihood of leaving. In contrast to peripheral users, the leave of core users considerably affects other core users that are part of the same cycle. This starts cascades of users leaving and thus destroys the core. Without the core, the periphery would not be able to sustain its reputation and would break down as well.

#### 2.2.6. Number and Length of Cycles

Unstable core-periphery structures can be avoided as long as *interlocking cycles* appear in the core. These contain users involved in multiple cycles which in turn receive a much higher reputation and increase the benefit (the relative reputation), also for others. Both the number of cycles in a network and their length have an impact on the largest eigenvalue $\lambda_{1}$ as illustrated in Appendix B. In general, we can conclude that $\lambda_{1}$
*increases* with the *number of cycles*, but *decreases* with the *length* of the cycle (keeping all other variables constant).

The number of cycles in the network further depends on the average density *m* (average number of links per user), a parameter discussed in the next section when we introduce the dynamics for the network.

### 2.3. Network Dynamics

In the previous section, we have explained that the reputation dynamics of Equation (Equation 2), for a *fixed network*, converges to an equilibrium state in which the relative reputations of users are fixed. This convergence time defines the time scale for the reputation dynamics. Dependent on their stationary reputation value that defines their benefit in relation to their costs τ, Equation (Equation 3), users can decide to leave the OSN, Equation (Equation 1), and will be replaced by new users joining the OSN. As described above, we assume that $N_{\mathrm{exit}}\left(T\right)=N_{\mathrm{entry}}\left(T\right)$, to keep *N* constant. *T* is the time at which entry and exit happen. We assume that the time scale for entry and exit, i.e., for changing the network structure, can be separated from the time scale of the reputation dynamics which is much shorter. This means that users make their decision based on the quasi-stationary benefit, which can only change after the network has changed.

If a user leaves the OSN, all her (incoming and outgoing) links are set to zero. The newcomer will establish the same number of incoming and outgoing links *on average* but, assuming that she does not know all other users, these links will be distributed at random. We assume that each user follows on average *m* other users. Precisely, there is a constant probability *p* that a new user links to the (N−1) other users, and m=p(N−1) is a constant related to the average density of a random OSN.

Our major interest in this paper is in the role of the cost τ that according to Equations (Equation 1), (Equation 3) defines the level at which users will leave the OSN, measured in terms of the relative reputation. The latter is between zero and one, so τ gives the fractional benefit that must be reached to stay in the OSN. In our computer simulations, we will vary this level from zero to 0.5 to study the impact of increasing costs.

τ=0 would imply no costs. To still allow for a network dynamics in this case, we apply the so-called extremal dynamics. In this case, we choose the user with the lowest relative reputation, and force her to leave the OSN, to allow a new user to enter. In case of several users with the same low *b* value, we choose one of them randomly. The other limiting case τ=1 would imply that all *N* users will leave and be replaced by a completely new cohort. Then, the network at every time *T* starts as a new random network, which has no chance to evolve. Hence, small or intermediate values of τ would be most appropriate.

## 3. Results of Computer Simulations

### 3.1. Performance

How should one measure the “performance” of an OSN? Users join the OSN for a purpose and, as we have explained in Section 2.1, here we assume that the benefits of users can be measured in terms of their *relative reputation*, which should be possibly increased. This implies that not only the number of followers is taken into account, but also their “value” in terms of their own reputation. Hence, it would be obvious to use the *long-term average* over all users’ relative reputation as a systemic measure whether or not the OSN fulfills the expectation of its users. This will serve as a proxy for the performance of the OSN.

Specifically, we build on the relative reputation, $b_{i}\left(T\right)$, Equation (Equation 3) of a user *i*, obtained in the quasi-stationary limit at time *T*. Then, both $X_{i}\left(t\right)$ and $X_{\max}\left(t\right)$ can be expressed by the corresponding values $X_{i}^{\lambda_{1}}\left(T\right)$ and $X_{\max}^{\lambda_{1}}\left(T\right)$ from the eigenvector $X^{\lambda_{1}}$ and we find for the average reputation of all *N* users
(12)$$\bar{b}\left(T\right)=\frac{1}{N}\sum_{i=1}^{N}b_{i}\left(T\right)=\frac{1}{N}\sum_{i=1}^{N}\frac{X_{i}^{\lambda_{1}}\left(T\right)}{X_{\max}^{\lambda_{1}}\left(T\right)}\;;\;\langle b\rangle =\frac{1}{R}\sum_{r=1}^{R}{\bar{b}}_{r}\left(T_{\max}\right)$$
$\bar{b}\left(T\right)$ refers to the population average at a given time *T* which can considerably fluctuate because of stochastic influences when changing the network structure at every time step *T*. Therefore, we define the *long-term average benefit*
$\langle b\rangle$ which is a system average taken over a large number of independent runs R=100, where the average satisfaction was measured after considerable long time $T_{\max}=12,000$.

The results of our simulations are shown in Figure 2 for various values of the costs τ and the average number of links *m*. The most remarkable observation, prominently shown on the left side of Figure 2, is the maximum average benefit ${\langle b\rangle }_{\max}$ for a non-zero cost at $\tau_{\max}=0.2$. Counterintuitively, this result implies that adding a cost for participating in an OSN maximizes the average reputation in the system.

The right part of Figure 2 demonstrates how this effect depends on the other important model parameter, the average number *m* of users, a new user tends to follow. Here, we clearly observe that for costs below $\tau_{\max}$, *m* has almost no influence on the performance, which is interesting enough because one would assume that a larger number of potential followers would always improve the situation. Note, however, that the performance relates to the *relative* reputation, i.e., $X_{i}$ may increase with *m* but so does $X_{\max}$. For costs above $\tau_{\max}$, we see a drastic decrease in the performance which depends on *m* in a much more pronounced way. In fact, there is a non-monotonous dependence—*increasing*
*m* in the range of small values will further *decrease* the performance.

To better understand these interesting results about the improvement of performance with increasing costs, we have to refer to the robustness of the OSN, which is inherently related to the stability of the core-periphery structure already discussed in Section 2.2.

### 3.2. Robustness

An OSN is said to be *robust* if the decision of users to leave the OSN will not trigger large cascades of further users leaving. Hence, we could use the sheer fraction of users remaining in the OSN at time *T*, $Y\left(T\right)=1-N_{\mathrm{exit}}\left(T\right)/N$, as a measure of its robustness. However, because this dropout is always compensated by a number of new users entering the OSN, we will need a different robustness measure.

We recall that the robustness of the OSN depends on the existence of a distinct *core-periphery* structure for which the stability conditions are explained in Section 2.2 and Appendix C. This core-periphery structure is challenged at every time *T* because of the entry and exit of users and the corresponding formation and deletion of links. Therefore, cascades of users leaving can generally not be avoided, as exemplified in Appendix C. However, such cascades should not destroy the whole OSN, in particular not the *core* of the network that contributes the most to the average benefit of its users and prevents the periphery from leaving the OSN.

Consequently, we will use the *lifetime*
$\Omega_{Q}$ of the core *Q* (measured in network time *T*) as a proxy for the robustness of the OSN. Because of considerable fluctuations, similar to performance, we will use the *average*
$\langle \Omega_{Q}\rangle$. In a similar fashion to performance, $\langle \Omega_{Q}\rangle$ is averaged over $T_{\max}=12,000$ network time steps, allowing for sufficient cycles of core creations and destructions.

Figure A1 in Appendix D illustrates, for a sample network, that the robustness measure $\langle \Omega_{Q}\rangle$ is maximized for the optimal cost level $\tau_{\max}=0.2$. Specifically, we notice a non-monotonous behavior. The lifetime of the core-periphery structure is mostly threatened by peripheral users attracting followers, without reciprocally contributing to the benefit of others (see Appendix C). A non-zero cost, $0<\tau <\tau_{\max}$, prevents this “behavior” to some extent, but a cost too high, $\tau_{\max}<\tau <1$, rather destroys the core and, hence, the depending periphery.

Because both performance and robustness are maximized for the optimal cost level $\tau_{\max}=0.2$, one could argue that performance is simply a substitute of robustness. This simplified explanation, however, does not hold. As Appendix C shows in detail, the core can be destroyed by cascades that involve core users. However, the replacement of users leaving by new users at each time step *T* also bears the chance that random rewiring leads to the *recovery* of the core, thanks to new users linked to the core.

That means, to maximize performance not only the *lifetime* of the core matters, but also the *recovery time*
$\Pi_{Q}$ it takes to rebuild the core. The OSN should recover quickly, to not affect the long-term benefit of its users. As Figure A1 in Appendix D shows for a sample network, the averaged recovery time $\langle \Pi_{Q}\rangle$ of the core-periphery structure becomes shorter if τ increases, i.e., if more new users enter the OSN, and new links increase the chance of establishing reciprocal relations. However, again, if the randomness associated with this process becomes too high, favorable structures may get destroyed. A small, but considerable cost $\tau_{\max}=0.2$ can balance these counteracting processes.

In conclusion, the cost value $\tau_{\max}$=0.2 optimizes the *ratio* between these two time spans, the average lifetime of the core, $\langle \Omega_{Q}\rangle$, and the average recovery time of the core, $\langle \Pi_{Q}\rangle$. It is only their combined impact of maximizing $\langle \Omega_{Q}\rangle$ while minimizing $\langle \Pi_{Q}\rangle$ that explains the maximum of performance $\langle b\rangle$, as discussed in detail in Appendix D.

### 3.3. Core Size and Largest Eigenvalue

In Section 2.2 we already shortly discussed how an increase in cost τ affects the structure of the OSN. In particular, the length of both simple chains and of cycles of followers will be reduced (Equation (Equation 11)). This results in a decreasing size *Q* of the core built by users that belong to one or more cycles. Figure 3 shows the histogram of the core sizes, P(Q) for two different costs, τ=0 and τ=0.25, where the first one is only used as a reference case. For each value of τ we ran R=100 simulations. The end of each simulation occurred at the time step $T_{\max}=12,000$ at which point we measured the core size and the largest eigenvalue of the corresponding adjacency matrix.

One recognizes that with increasing cost, the distribution gets more skewed, with its maximum shifted to smaller values of *Q*. To allow for a real comparison of the different simulations, we have taken into account only those realizations where the core-periphery structure encompasses the whole network (i.e., one connected component, and no isolated users, or groups of users). Figure A2 shown in Appendix E also illustrates the structure of the resulting OSN. One clearly observes that with increasing costs there is a tendency of users in the core to follow more other users.

The second structural insight comes from the histogram of the largest eigenvalue, $P(\lambda_{1})$, shown in the right part of Figure 3. Compared to the case with no costs, we observe a considerably broader distribution, with the maximum shifted to a larger value of $\lambda_{1}$. The latter indicates that with increasing costs there is a much larger likelihood to find larger values of $\lambda_{1}$. This, in turn, implies shorter simple chains and shorter cycles. In line with the argumentation in Appendix B, we can also confirm that larger $\lambda_{1}$ correspond to more cycles inside the core, and thus larger average reputation, which can be verified by comparing the snapshots of Figure A2.

We remind that the condition (Equation 11) only refers to simple cycles and *simple* chains, i.e., chains in which each user has exactly one follower. If users have several followers, their benefit-to-cost difference can overcome the condition (Equation 11) even if their position in the chain or in the cycle would not allow for this. In addition, such users automatically boost the reputation of all other users downstream. As the cost τ increases, having more than one follower becomes crucial in particular for peripheric users to stay. Having more than one follower, on the other hand, also increases the chance of creating new cycles, which in turn increases $\lambda_{1}$. This again feeds back both on the core and the periphery of the OSN, increasing the pressure towards a more compact core and shorter chains. Hence, it is in fact the relation with $\lambda_{1}$ that facilitates the role of the cost τ in shaping the network.

## 4. Discussion

In this paper, we have extended previous works on online social networks (OSN) in different respect: (a) we have studied directed networks of users and their followers; (b) users are characterized by a dynamic variable, their reputation, which is quantified not by simple topological centrality measures, but by a non-local measure, eigenvector centrality; (c) to model changes in reputation, we use the idea of hetero-catalytic processes, i.e., cycle-related feedback, which is not often used in describing the dynamics of online social networks. These features are reflected in an agent-based model, which captures several generic aspects of OSN. We summarize some of these in the following.

### 4.1. Core-Periphery Structure

Most OSNs are characterized by a core of active users that are closely linked to each other, and a periphery of less active users that are loosely connected to the core. Empirical studies have shown [2,21] that the core of OSN is not only quite large, compared to the size of the OSN, but also “deep”, i.e., most users are well integrated in the OSN. Our model can reproduce such a structure, but also to explain its origin, as a combination of direct and indirect reciprocity. The former implies mutual directed links between two users, whereas the latter occurs if users are part of closed cycles and thus benefit from others through the cycle.

### 4.2. Reputation Dynamics

Many models of OSN take the number of “friends”, i.e., direct links between users into account and take degree and traditional centrality metrics as proxies of a user’s reputation [22,23,24]. Our model, in contrast, not only considers that such social relations are *directional* (i.e., a link from user *j* to *i* does not necessarily implies a link from *i* to *j*), but also explicitly takes the different impact of users into account, weighting it by means of a (generalized) *reputation*. Notably, reputation is not determined by the user herself, instead it results from the interaction with others. A user is said to have a high reputation if it is followed by many other users with high reputation. In this way, the *value* or "quality" of one’s followers is taken into account, in addition to the sheer *number*. Reputation that is not actively maintained will fade out in the course of time, so that users that are not able to attract followers automatically lose in impact.

### 4.3. Cost-Benefit Analysis

In our model, users join the OSN for a purpose. That means they bear costs, to obtain some benefit. If their costs are higher than their benefits, users *leave* the OSN. Hence, our model includes an *entry-exit dynamics* which is motivated by rational arguments. We have assumed that a user’s benefit increases with the number of followers it attracted, which in turn boost its reputation. Hence, benefits are measured by the relative reputation of a user, scaled by the maximum reputation in the OSN. The latter allows us to better compare OSNs of different sizes and link densities. Costs, on the other hand, are assumed to be fixed, they include for example the effort of using the website or to maintain a profile.

Our main focus in this paper was on the role of the *cost*
τ incurred for every active user. Increasing τ would imply a pressure on users to leave because it worsens their cost-benefit relation. This short-term negative effect, however, has a positive influence on the core-periphery structure of the OSN. As we demonstrate in this paper, an optimal cost level can increase the *long-term benefit* of users in terms of a higher relative reputation.

### 4.4. Performance

To estimate the impact of increasing costs, we have defined the long-term benefit averaged over the whole OSN, $\langle b\rangle$, as a performance measure. Intuition would suggest that $\langle b\rangle$ monotonously decreases with increasing costs up to a point where the whole OSN collapses. Interestingly, this picture does not hold for comparably small cost values. On the contrary, a small cost up to $\tau_{\max}=0.2$ improves the situation, i.e., the long-term benefit *increases* compared to a reference case without any cost (where no user would leave). In other words, a small cost forces those users to leave which were never able to attract any follower. This, in our model, gives way to new users that might be more successful in this respect.

### 4.5. Robustness

As a second important insight, we analyze both mathematically and by means of computer simulations how increasing costs change the structure, and hence the robustness, of the OSN. Our model is set up in a way that it allows the expression of the outcome of the reputation dynamics by $\lambda_{1}$, the largest eigenvalue of the adjacency matrix that describes the interaction between users. We verified that an increasing cost τ leads to higher values of $\lambda_{1}$, which means a smaller, but more compact core and, most remarkably, in an increasing likelihood to have more than one follower. This, on the other hand, *increases* the direct and indirect *reciprocity* that characterizes the core.

Robustness implies that cascades of users leaving will not destroy the core, i.e., the lifetime of the core can be used as a proxy measure of robustness. We have shown that this lifetime is maximized for the optimal cost value $\tau_{\max}=0.2$.

We emphasize that the above conclusions are obtained from a model that, as with every model, only captures part of the features of real OSN. The value of the model is in what it can produce *despite* some of the simplified assumptions. Our insights now allow us to specifically search for optimal cost-benefit relations, or to test other assumptions to calculate the benefit.

Why is this issue important? First, there *are* costs involved in being a member of an OSN, even if they are implicit. These include, among others, the efforts to login and to retain one’s own social network (e.g., by maintaining regular information stream), the effort of learning how to efficiently use the GUI of the provider, or the effort to adapt to changes in the GUI. The latter can make it harder to maintain social contacts, at least temporarily. As Garcia et al. [2] argued, it was ill-timed interface changes that caused the massive dropout of users from Friendster, which came at a time when new competitors, such as Facebook, were ramping up in popularity.

There is a second issue involved in this discussion. Even if most users enjoy participating in an OSN free of charge, companies would like to know these users would respond, if at some point in time costs such as monthly membership fees are introduced. Will members remain loyal, or will this lead to a massive exodus of users, making the OSN less attractive for investors? Subsequently, if users leave how would this affect the OSN? What will be the impact of less active users leaving compared to core users leaving? What incentives should be introduced to keep power users engaged?

With our agent-based modeling framework that consider both entry-exit dynamics and cost-benefit considerations of users, we can address such questions. It also helps to better understand the relation between the integration of users and the overall performance of the OSN, which allows to the formal consideration of issues such as user satisfaction and service quality.

## Figures and Tables

**Figure 1 entropy-22-01073-f001:**
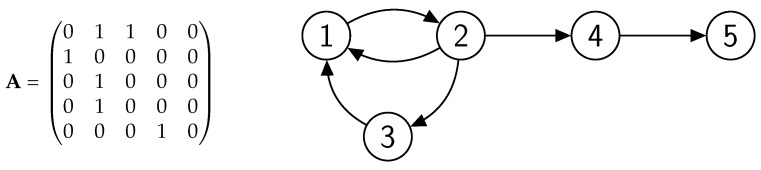
Sketch of an OSN (**right**) that displays a core-periphery structure (see text) and the corresponding adjacency matrix A (**left**).

**Figure 2 entropy-22-01073-f002:**
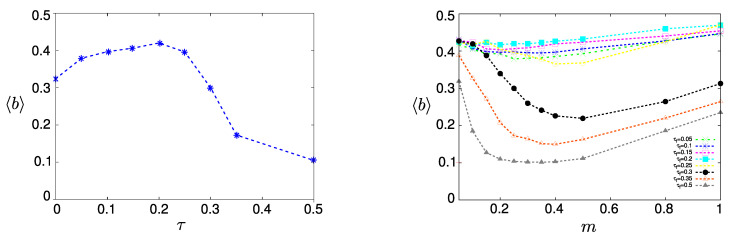
Long-term average benefit $\langle b\rangle$, Equation (Equation 12) for varying costs τ (**left**) and average number of links *m* (**right**). Other parameters m=0.25, N=100, $T_{\max}=12,000$.

**Figure 3 entropy-22-01073-f003:**
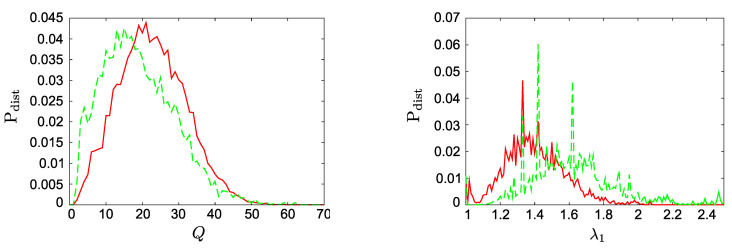
(**left**) Distribution P(Q) of core sizes *Q*, (**right**) Distribution $P(\lambda_{1})$ of the largest eigenvalue $\lambda_{1}$. (red) τ=0, (green τ=0.25). Other parameters N=100, m=0.25, $T_{\max}=12,000$.

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
