# Peer review of "Modeling User Reputation in Online Social Networks: The Role of Costs, Benefits, and Reciprocity"

_entropy, 2020, doi:10.3390/e22101073_

Round 1
Reviewer 1 Report
In this work, the authors present a simple but very interesting agent-based model social network dynamics. They use this model to explore the role of cost and reputation on the robustness of the social network core. They show that there exists a non-zero value of the cost parameter for which they obtain the highest average reputation and the most significant and most robust core. While very simple, this model represents a base for developing more sophisticated tools to study the life and death of online social networks.
The paper falls within the scope of the Entropy journal and special topic, Dynamic Processes on Complex Networks. However, few things need to be clarified and corrected before it can be considered suitable for publishing.
1) New links are introduced only between new users and old users, never between two old users. The number of outgoing links per user is thus m=p(N-1) and, on average, is the same. The number of incoming links depends on the node's age, but this dependence is not trivial or obvious.
The first conclusion is that the network is random, without hubs or any other inhomogeneities, including core-periphery structure, confirmed by Figure A2. The authors should elaborate on the network dynamics and the structure of resulting networks in more detail.
2) p is a probability that the newly added node will connect to an old node. m is the average number of nodes that a new node will follow. m is not bounded from above. m smaller than one means that a new node will attach to less than one node on average. These networks are spars and do not have many cycles. Why did the authors decide to only consider networks with m<=1? How this influences the results and the conclusions in the paper?
3) As one of the factors that authors consider is reciprocity. As the authors notice, it can be direct and indirect. The described model does not enable direct reciprocity. A new node may follow an old one, but the old one will never follow back since linking rules do not enable this. It can only happen in the starting network, and it will be sporadic for m<=1. The authors should emphasize this in the paper.
There are a few technical errors that should be corrected.
There is an occurring error trough the whole manuscript. Before phrase i.e. authors put "." instead of ",". The error occurs in the following lines: 44, 94, 222, 233, 321, 353, 355, 511,
In line 399 Frienster should be Friendster.
Please fill in Author Contributions, Funding, Acknowledgments, and Conflicts of Interest.
In lines 282 and 283, authors say, "use the average 〈Ω_{Q}〉 taken over a considerable long time T_{max} (and possibly also averaged over a large number R of independent runs)". Do authors average over a large number R of independent runs or not? Please clarify and correct this.
Author Response
*1) New links are introduced only between new users and old users, never between two old users. The number of outgoing links per user is thus m=p(N-1) and, on average, is the same. The number of incoming links depends on the node's age, but this dependence is not trivial or obvious.
The first conclusion is that the network is random, without hubs or any other inhomogeneities, including core-periphery structure, confirmed by Figure A2. The authors should elaborate on the network dynamics and the structure of resulting networks in more detail.
------------
The reviewer is correct that a node's age is related to the number of followers that this node had been able to attract in a non-trivial way. For example, a well-embedded core node will rarely get removed by the entry/exit dynamics, due to its high reputation, and will have more chances to get incoming connections from new nodes. At least until the core it belongs to is destroyed. On the other hand, the new nodes which have connected to the core node are also more likely to get removed by the entry/exit dynamics, thus in turn reducing the number of followers of the code node.
However, as we outline through the paper a recurrent emergent structure in the network is the core-periphery. Moreover, this structure invariably emerges even when the dynamics start from a random network and evolve via simple random re-wiring. This speaks to the fundamental nature of core-periphery. In fact, Figure A2 actually confirms that by showing an emergent core-periphery structure in all snapshots. It is precisely that structure, which we are trying to understand, and especially its interplay with the cost of participation. In particular, we investigate the break-down of the core periphery structure with a minimal example in Appendix C and the exact structure of the core and the periphery in Section 2.2.
*2) p is a probability that the newly added node will connect to an old node. m is the average number of nodes that a new node will follow. m is not bounded from above. m smaller than one means that a new node will attach to less than one node on average. These networks are spars and do not have many cycles. Why did the authors decide to only consider networks with m<=1? How this influences the results and the conclusions in the paper?
---------------
There is no fundamental reason for choosing m<1. Larger values for m lead to denser networks and much faster emergence of a core. Moreover, such cores will tend to exhibit more cycles in them. With all investigations we did for m>1 we noticed that the results pertaining to resilience and average benefit (Figure A1) do not change qualitatively. Rather the scale for the average long-term benefit, average lifetime of the core and average time for the core to recover change quantitatively.
3) As one of the factors that authors consider is reciprocity. As the authors notice, it can be direct and indirect. The described model does not enable direct reciprocity. A new node may follow an old one, but the old one will never follow back since linking rules do not enable this. It can only happen in the starting network, and it will be sporadic for m<=1. The authors should emphasize this in the paper.
-----------------
Thank you for your comments, we clarified the point in our paper.
In fact, the discussion about reciprocity was more for motivational purposes. But it is interesting to note that even though our model relies on the weaker indirect reciprocity it is still enough to reproduce the core-periphery structure.
* There is an occurring error trough the whole manuscript. Before phrase i.e. authors put "." instead of ",". The error occurs in the following lines: 44, 94, 222, 233, 321, 353, 355, 511,
-----------------
We thank the reviewer for spotting this. All mentions of i.e., have been corrected.
* In line 399 Frienster should be Friendster.
-----------------
We thank the reviewer for spotting these typos. They have been corrected.
* Please fill in Author Contributions, Funding, Acknowledgments, and Conflicts of Interest.
-----------------
We did so.
* In lines 282 and 283, authors say, "use the average 〈Ω_{Q}〉 taken over a considerable long time T_{max} (and possibly also averaged over a large number R of independent runs)". Do authors average over a large number R of independent runs or not? Please clarify and correct this.
-----------------
We have removed the confusing part about possible number of independent runs. In fact, the average is taken only in time. In addition, we have expanded Appendix D with details on how the average lifetime and average time to create a new core are measured.
Reviewer 2 Report
The authors introduce a dynamical model of networks with a replacement dynamic
driven by "reputation" --- effectively eigenvector centrality. They perform a
numerical analysis of this model and investigate the impact of replacements on
long-term centrality distribution. They motivate the model from the
perspective of online social network analysis.
While I appreciate the pedagogical examples of eigenvalue problems made by the
authors, I'm not sure that this manuscript should be published in Entropy.
First, a significant partition of the manuscript goes over known results,
although the connection is not made explicit. Indeed, the reputation the
authors derive is just the eigenvector centrality, a well-known measure of
centrality, usually derived starting from Eq. (7) directly; see, for example,
"Networks" by Mark Newman (Oxford University Press 2018). The sequence of
benefit->reputation->normalized reputation obscures this correspondence.
Second, I think that the model is ad-hoc and unlikely to say anything of
substance about a real online social network. Here are examples of question
that come to mind: Why should "reputation" be exponentially hard to sustain?
Why not a positive feedback loop? Why would one leave if they have a low
reputation? Are there not essential mechanisms unrelated to network ties that
can drive reputation?
The rules chosen are not grounded in any empirical study of how OSNs evolve,
and I find the model chosen by the authors hard to believe. As a result, I
don't think we can draw any conclusion about real OSN from the simulations,
even if we focus only on the notion of reputation---the model is just too
crude.
Third and relatedly, the analogies with OSN are thin at best. What is the
network in the case of amazon? Or youtube? Reputation is mediated by ties in
the proposed model, but the reputation is derived from exposure to content and
comments on Youtube.com. Amazon doesn't function as a social network at all.
Reddit karma is independent of "ties" and rewards interesting posts. The only
platforms that could plausibly support a reputation dynamic that looks a bit
like the model are Twitter, Facebook, and Friendster.
To be clear: It is fine to study a model for its own sake. All network models
are highly stylized. In this case, however, given the claimed relevance to
actual OSN I would suggest to either support the mechanisms much more clearly
and empirically, or to lean into the fact that this is a toy-model. I would
also clarify the connection with the well-known eigenvector centrality.
There are also a few minor issues:
* Shouldn't there be a time lag in Eq (1)? The agents update on information at
time t-1.
* The manner in which the Perron-Frobenius is used is confusing. The theorem
holds for non-negative matrix only when they are irreducible. Hence, there's
no such thing as a "Perron-Frobenius" eigenvalue of the digraph if it is not
strongly connected. The example shown in Fig. 1 is not irreducible---hence
the theorem as normally understood doesn't hold in this case. One has to
focus on the strongly connected component.
* There are insufficient details on the numerical experiments in Fig.2. What
are N, T_max, and m in panel a?
* L135 m is implicitly defined as the "maximizing component," but z is used in
the rest of the text.
* We have insufficient information on how the life-time of the core is
measured. How do you define the core when there are separate strongly
connected components? What counts as the birth of a core and its death?
* Fig.3: We don't know whether the model is ergodic. With this in mind, are
histogram computed over a single run of the model or over multiple
realizations?
* Fig.3 again: What if one compared tau=0.25 to 0 < tau < 0.25 . Tau=0 is the
special case where one node is removed at a time, a "singular point" of the
dynamics.
Some typos:
* L20 Loosing -> losing
* L21 Incomplete sentence. It can be fixed as: "As a recent empirical study
[11] has shown that the dropout" -> "As a recent empirical study [11] has
shown, the dropout" (is 2013 "recent"?)
* L36 Incomplete sentence. "This implies to consider (i).." Perhaps "This
implies we should consider"
* L44 "I.e. models" -> "For example," (i.e. means "That is")
* L355 loose -> lose
* L133, L444. Rendering issue for eigenvectors notation.
* Fig A2: Could be an artifact of the layout, but there's no "left" "middle"
and "right". It should be something like top-left, top-right, and bottom.
* References 16 and 17 are identical.
Author Response
* First, a significant partition of the manuscript goes over known results,
although the connection is not made explicit. Indeed, the reputation the
authors derive is just the eigenvector centrality, a well-known measure of
centrality, usually derived starting from Eq. (7) directly; see, for example,
"Networks" by Mark Newman (Oxford University Press 2018). The sequence of
benefit->reputation->normalized reputation obscures this correspondence.
Second, I think that the model is ad-hoc and unlikely to say anything of
substance about a real online social network. Here are examples of question
that come to mind: Why should "reputation" be exponentially hard to sustain?
Why not a positive feedback loop? Why would one leave if they have a low
reputation? Are there not essential mechanisms unrelated to network ties that
can drive reputation?
The rules chosen are not grounded in any empirical study of how OSNs evolve,
and I find the model chosen by the authors hard to believe. As a result, I
don't think we can draw any conclusion about real OSN from the simulations,
even if we focus only on the notion of reputation---the model is just too
crude.
Third and relatedly, the analogies with OSN are thin at best. What is the
network in the case of amazon? Or youtube? Reputation is mediated by ties in
the proposed model, but the reputation is derived from exposure to content and
comments on Youtube.com. Amazon doesn't function as a social network at all.
Reddit karma is independent of "ties" and rewards interesting posts. The only
platforms that could plausibly support a reputation dynamic that looks a bit
like the model are Twitter, Facebook, and Friendster.
To be clear: It is fine to study a model for its own sake. All network models
are highly stylized. In this case, however, given the claimed relevance to
actual OSN I would suggest to either support the mechanisms much more clearly
and empirically, or to lean into the fact that this is a toy-model. I would
also clarify the connection with the well-known eigenvector centrality.
---------
The reviewer's main complaint seems to be related to the notion of reputation and the mechanisms which we chose to model it. While it is true that our model uses a crude proxy for reputation, namely the sheer number of one's followers, reputation itself is an excellent first approximation for the benefit that a user receives from an OSN. Reputation increases the influence of one's content, as measured by the likelihood that this content will be acknowledged, consumed and re-distributed in the community.
This is important not only for private individuals, but also for commercial entities which benefit directly from delivering content to millions of users. Individual reputation is, therefore, a key component in the benefit of participation.
We actually agree with the reviewer that measuring reputation as eigenvector centrality does not reflect the precise ways that most OSNs build up user reputation, especially in cases where direct follower relationship is not designed into the OSN. We comment, however, that in the latter cases (such as Amazon or Youtube) reputation is perceived on an aggregate level, which actually makes it even more important for the user to maintain it. For example a Youtube content creator needs to maintain a steady stream of quality content in order to maintain his reputation, measured as number of views or likes/dislikes ratio. Compared to a Facebook user, the net benefit of the Youtube content creator decays much more rapidly in case he is inactive.
However, we would like to point out that the main focus of our paper is on investigating the interplay between the resilience of an OSN and the cost of participation in it.
As every modelling approach, ours also only captures a part of reality. Our model certainly does
not include all features of a real OSN, and even simplifies those that
are included. For example, we ignored the component of the benefit that
users obtain from consuming content, and only focused on the
contributions from the ability to capture attention,
i.e. reputation. Though justified, this simplification certainly leaves out
other interesting insights. Another example is our entry/exit
dynamics. Arguably, new users do not form connections completely at
random, but may decide based on the reputation of existing users
(in a multiplicative process akin to preferential attachment) or based on existing offline
relationships.
In choosing the right amount of complexity we
were driven by one main consideration - what is the minimum amount of
model complexity needed to study the interplay between cost of usage and
resilience?. This is an important question, because there is always a
trade-off between complexity and tractability. It is surprising to find out that we do not need complex assumptions about user engagement, user entry or benefit function.
Already accounting for user reputation, as recursively dependent on the reputation
of one's followers, and including a rudimentary network dynamics allows
us to recover a prevalent structure of real OSNs (core-periphery) and to demonstrate that there exists an optimal cost of usage for an OSN,
which maximizes resilience. That this result arises from a simple model
speaks only about its fundamental nature. Therefore our main claim is that a positive cost of usage maximizes resilience and that this is a general result.
* Shouldn't there be a time lag in Eq (1)? The agents update on information at
time t-1.
-----------------
We note that our approach in modelling the network dynamics of an OSN relies on two timescales. The first is a fast timescale during which user reputations converge to their steady states. We provide an analytical method based on the Perron-Frobenius theorem to compute the steady states.
Eq. (1) refers to the second timescale, which is slower and reflects the entry/exit dynamics.
At the end of each time step, t, on that timescale, a user has a steady-state benefit, computes his net benefit by subtracting costs and decides to stay or leave the OSN. All of these actions occur simultaneously on this slow timescale. Therefore, a lag in Eq. (1) is not necessary.
* The manner in which the Perron-Frobenius is used is confusing. The theorem
holds for non-negative matrix only when they are irreducible. Hence, there's
no such thing as a "Perron-Frobenius" eigenvalue of the digraph if it is not
strongly connected. The example shown in Fig. 1 is not irreducible---hence
the theorem as normally understood doesn't hold in this case. One has to
focus on the strongly connected component.
-----------------
Indeed, the reviewer makes a good technical point. In case of a reducible, non-negative matrix, the Peroon-Frobenius theorem does not guarantee that the maximum eigenvalue of the adjacency matrix will be strictly positive, nor that the corresponding eigenvector will be strictly positive.
Instead both the eigenvalue and eigenvector will be at least non-negative. The eigenvalue, in turn, may not be unique. For the entry/exit dynamics we propose, however, a non-negative eigenvector simply implies that some users may end with a zero reputation at the end of each network time step. For example, a user who, by chance, started with no followers will see his positive initial reputation decay quickly to zero. Such users will be removed by the entry/exit dynamics preferentially.
Nevertheless, it is true that the Peroon-Frobenius theorem cannot be applied in its original form. To avoid confusion we have removed the reference to it, as our analysis does not critically depend on it.
* There are insufficient details on the numerical experiments in Fig.2. What
are N, T_max, and m in panel a?
-----------------
We thank the reviewer for noticing this omission. The corresponding captions have been expanded with this missing information.
* L135 m is implicitly defined as the "maximizing component," but z is used in
the rest of the text.
-----------------
We thank the reviewer for spotting this typo. Indeed 'z' must be used instead of 'm'.
* We have insufficient information on how the life-time of the core is
measured. How do you define the core when there are separate strongly
connected components? What counts as the birth of a core and its death?
-----------------
The reviewer is correct that measuring the life-time of the core could be explained better. We have added extra information on that in Appendix D.
As for the definition of the core, we refer the reviewer to Section 2.2. Core-periphery structure, where a core is defined precisely as "the largest strongly connected component (SCC) in the OSN", which is by definition unique.
* Fig.3: We don't know whether the model is ergodic. With this in mind, are
histogram computed over a single run of the model or over multiple
realizations?
-----------------
We have added explanations about the computation of the histograms in the main text referring to the figure. We thank the reviewer for this suggestion.
* Fig.3 again: What if one compared tau=0.25 to 0 < tau < 0.25 . Tau=0 is the
special case where one node is removed at a time, a "singular point" of the
dynamics.
-----------------
According to the discussions in Section 3.3 and Appendix B, we expect that increasing costs leads to a smaller core and higher likelihood of larger \lambda_1. While Figure 3 indeed compares tau=0 vs tau=0.25, Figure A.2 also adds tau=0.2 vs tau=0.3. The conclusions regarding core size remain the same. As mentioned in Section 3.3, higher costs will result in more short cycles being added to the core and shorted simple chains in the periphery.
The overall conclusion is that as long as there is a stable core (i.e., tau is not too high) increasing tau consolidates the core users into a smaller strongly connected component, which however exhibits more cycles in it. This dynamics actually increases the average reputation of the core users up to the point of optimal costs. Beyond that the core is increasingly more likely to get affected as simple chains coming out of it become less and less likely.
Some typos:
* L20 Loosing -> losing
* L21 Incomplete sentence. It can be fixed as: "As a recent empirical study
[11] has shown that the dropout" -> "As a recent empirical study [11] has
shown, the dropout" (is 2013 "recent"?)
* L36 Incomplete sentence. "This implies to consider (i).." Perhaps "This
implies we should consider"
* L44 "I.e. models" -> "For example," (i.e. means "That is")
* L355 loose -> lose
* L133, L444. Rendering issue for eigenvectors notation.
* Fig A2: Could be an artifact of the layout, but there's no "left" "middle"
and "right". It should be something like top-left, top-right, and bottom.
* References 16 and 17 are identical.
-----------------
We have also corrected all of these typos.
Round 2
Reviewer 1 Report
The authors have addressed most of my comments and remarks. The manuscript is about a simple model that analyses one possible mechanism that may drive the emergence of core-periphery structures in online social networks. As such, it provides an essential piece of knowledge necessery for understanding a much more complex behavior of online social networks. I find this work relevant for the scientific community interested in the dynamics of social networks' dynamics, including online, and recommend it for publication.
Reviewer 2 Report
The authors fixed minor technical issues like an inappropriate use of the Perron-Froebinus theorem and typos. Still, they did not address my main concerns, either in the paper or their responses.
For example, the connection with eigenvector centrality is only mentioned in passing, in their conclusion.
Crude modeling choices --- like the exponential difficulty of maintaining "reputation" or the separation of time-scales --- are still not justified.
Thin analogies with platforms like Amazon or Youtube, and reddit are still present.
Again, I wholly agree that simplifications are necessary when modeling, and I also personally enjoy modeling for its own sake.
However, if the authors wish to present their results as relevant to OSNs, then empirical motivations should be a pre-requisite.
As actionable recommendations go, I believe that making the connection with eigenvector centrality explicit is necessary for the sake of intellectual honesty.
I suppose the paper won't be incorrect if the other comments are not addressed.
But I still believe that the results would be better served by a presentation that (i) stems from strong empirical motivations or (ii) is fully decoupled from empirical systems like OSNs, and studied as a toy-model for its own sake.
Some further typos
* l174 adjacenny
* change of verb tense mid-sentence in the discussion (goes from present perfect to present) in c)